# Dietary Data in the Malmö Offspring Study–Reproducibility, Method Comparison and Validation against Objective Biomarkers

**DOI:** 10.3390/nu13051579

**Published:** 2021-05-09

**Authors:** Sophie Hellstrand, Filip Ottosson, Einar Smith, Louise Brunkwall, Stina Ramne, Emily Sonestedt, Peter M. Nilsson, Olle Melander, Marju Orho-Melander, Ulrika Ericson

**Affiliations:** 1Department of Clinical Sciences Malmö, Diabetes and Cardiovascular Disease-Genetic Epidemiology, Lund University, 205 02 Malmö, Sweden; louise.brunkwall@med.lu.se (L.B.); marju.orho-melander@med.lu.se (M.O.-M.); ulrika.ericson@med.lu.se (U.E.); 2Department of Clinical Sciences Malmö, Cardiovascular Research, Hypertension, Lund University, 205 02 Malmö, Sweden; filip.ottosson@med.lu.se (F.O.); einar.smith@med.lu.se (E.S.); olle.melander@med.lu.se (O.M.); 3Department of Clinical Sciences Malmö, Nutritional Epidemiology, Lund University, 205 02 Malmö, Sweden; stina.ramne@med.lu.se (S.R.); emily.sonestedt@med.lu.se (E.S.); 4Department of Internal Medicine, Skåne University Hospital, Lund University, 205 02 Malmö, Sweden; peter.nilsson@med.lu.se

**Keywords:** food intake, dietary assessment methods, reproducibility, validation, biomarker, fish, vegetables, fruits, citrus

## Abstract

Irregular dietary intakes impairs estimations from food records. Biomarkers and method combinations can be used to improve estimates. Our aim was to examine reproducibility from two assessment methods, compare them, and validate intakes against objective biomarkers. We used the Malmö Offspring Study (55% women, 18–71 y) with data from a 4-day food record (4DFR) and a short food frequency questionnaire (SFFQ) to compare (1) repeated intakes (*n* = 180), (2) intakes from 4DFR and SFFQ (*n* = 1601), and (3) intakes of fatty fish, fruits and vegetables, and citrus with plasma biomarkers (*n* = 1433) (3-carboxy-4-methyl-5-propyl-2-furanpropanoic acid [CMPF], β-carotene and proline betaine). We also combined 4DFR and SFFQ estimates using principal component analysis (PCA). Moderate correlations were seen between repeated intakes (4DFR median ρ = 0.41, SFFQ median ρ = 0.59) although lower for specific 4DFR-items, especially fatty/lean fish (ρ ≤ 0.08). Between-method correlations (median ρ = 0.33) were higher for intakes of overall food groups compared to specific foods. PCA scores for citrus (proline betaine ρ = 0.53) and fruits and vegetables (β-carotene: ρ = 0.39) showed the highest biomarker correlations, whereas fatty fish intake from the SFFQ per se showed the highest correlation with CMPF (ρ = 0.46). To conclude, the reproducibility of SFFQ data was superior to 4DFR data regarding irregularly consumed foods. Method combination could slightly improve fruit and vegetable estimates, whereas SFFQ data gave most valid fatty fish intake.

## 1. Introduction

A significant part of chronic diseases can be prevented by leading a healthy lifestyle, including diet. Consequently, there is a need for improved understanding of the role of dietary intakes in disease prevention. However, dietary intake in epidemiological studies mainly relies on self-reported information, and all dietary assessment methods are prone to errors [1]. Irregular consumption of foods makes it difficult to remember and report dietary intake, which complicates valid assessment of long-term habitual intakes [2]. Previous results from the Malmö Diet and Cancer cohort indicate that the validity and reproducibility of intake assessments of some specific foods consumed on a non-regular basis, such as fish, are quite low [3,4,5,6]. This points towards some intakes being challenging to capture, and it may therefore be valuable to combine dietary assessment methods with different strengths and weaknesses in order to improve the ability to capture habitual dietary intake. In addition, biomarkers of dietary intakes can be important complements to self-reported dietary data and have been used to validate dietary assessment methods [7,8]. The biomarker 3-carboxy-4-methyl-5-propyl-2-furanpropanoic acid (CMPF), measured in human plasma, has previously been associated with dietary fish intake, especially fatty fish and fish oils [9,10,11]. Plasma β-carotene is an objective biomarker for fruit and vegetable intake [12,13], whereas proline betaine, which is present in citrus, has been identified as an objective biomarker of citrus intake [14,15].

In this study, we compared intakes from the two different dietary assessment methods used in Malmö Offspring Study (MOS): a 4-day food record (4DFR) and a short food frequency questionnaire (SFFQ). In a subsample, we also examined the reproducibility of data obtained by the two methods using repeated intake measurements with a mean time interval of 1.6 y. Finally, we examined the validity of data on fatty fish, fruit and vegetable and citrus intakes from each assessment method, as well as of data obtained by combining intakes obtained from the 4DFR and SFFQ, using the plasma biomarkers CMPF, β-carotene and proline betaine. This study evaluates the quality of dietary data in MOS, and may indicate which data to use regarding different foods, in order to capture dietary intake most accurately when further examining associations between diet and chronic disease.

## 2. Materials and Methods

### 2.1. Data Collection

MOS is an ongoing population-based cohort study where children and grandchildren (aged > 18 years) of the Malmö Diet and Cancer–Cardiovascular Cohort are recruited [16,17,18]. The participants visited the research clinic twice. At the first visit, venous blood was drawn after an overnight fast; anthropometrics were measured and the participants were instructed as to how to record the 4DFR (starting the day after the first visit) and how to fill in a SFFQ and a comprehensive questionnaire on other lifestyle and socioeconomic factors. All participants provided written informed consent and the Regional Ethics Committee of Lund University approved the MOS study protocols (Dnr: 2012/594).

### 2.2. Study Sample

From the start of the study in March 2013 until April 2017, 2644 individuals participated in baseline examinations (47% of the eligible participants). Among those, 1601 participants (54% women) completed both a 4DFR and SFFQ on selected foods and constituted the study sample for dietary method comparisons (Figure 1).

The participants that completed a 4FDR between 31 May 2014 to 13 June 2015 (*n* = 400) were invited to repeat the 4FDR and the SFFQ. The study sample for this reproducibility study included the 180 participants with complete repeated measurements from both 4DFR and SFFQ. The mean interval between the assessments was 1.6 years (±0.3).

Plasma metabolite levels were measured in 1433 out of the 1601 participants with complete dietary data from the baseline measurements and constituted the study sample for validation of citrus intake against proline betaine in plasma. When validating fatty fish intake against CMPF, 101 participants were excluded due to reported use of omega-3 fatty acid containing supplements during the 4FDR, leaving 1332 participants. Finally, for validation of fruit and vegetable intake against β-carotene, 132 of the 1433 participants were excluded due to reported use of multivitamin dietary supplements commonly containing β-carotene. Therefore, 1301 participants constituted the study sample.

### 2.3. Dietary Data

Dietary intake was assessed with a web-based 4DFR, Riksmaten2010, developed by the Swedish National Food Agency [19], and a semiquantitative SFFQ, developed by nutritionists working with MOS. The participants were instructed via a video (https://www.youtube.com/watch?v=DB3bzD0FJMg, accessed on 3 October 2013) and asked to record all they ate and drank during four consecutive days and to estimate their usual portion sizes using a booklet containing 24 photographs or household measurement (e.g., cups, spoons, deciliters, etc.). Each set of photographs showed different portion sizes with 5–9 options depending on dish/food item. The participants started the 4DFR one day after the first visit to the research clinic, a design chosen to make sure all weekdays were represented in the study and that all participants had at least one weekend day included in their 4DFR. For the repeated 4DFR, the participants were asked to start recording on the weekday following the last weekday of their first 4DFR.

The relative validity of the Riksmaten2010 was validated by comparing the total energy expenditure (TEE) measure by the objective double-labeled water technique to the reported energy intake (r = 0.40) [19]. The average daily food intake (g/d) was calculated based on information from the 4DFR and converted into nutrient and energy intakes (including alcohol) using the National food database “Riksmaten vuxna 2010” (in Swedish) version 10-05-05.

The SFFQ questionnaire included 32 selected food items (focusing on bread, vegetables, fruits, fish, and sources of fat in cooking, see Appendix A), three questions about beverages, and three about use of food replacement products (e.g., different shakes such as Nutrilett), i.e., intakes that may be consumed irregularly or seldom and thereby not satisfactorily captured when recorded during too few days as in a 4DFR. In addition, four questions about meal type, one question about use of probiotics, and a final question about previous substantial change of dietary habits were included. The participants were asked to indicate average intake frequencies during the last six months (eight alternatives, from “seldom/never” to “more than once per day” day). Additionally, fish portion sizes were asked for using a set of photographs with six different portion sizes. The SFFQ has not previously been validated.

### 2.4. Anthropometric Measurements

Height (m) was measured to the nearest centimeter, without shoes and hats. Weight (kg) was measured in light clothing on a calibrated balance beam or digital scale. Thereafter, body mass index (BMI; kg/m^2^) was calculated from these measurements.

### 2.5. Other Variables

Physical activity levels (PAL) were based on two questions about physical activity at work and leisure-time physical activity (LTPA) (both on a four-level scale ranging from sedentary to heavy manual labor/exercise ≥ 3 × 30 min/week) in the 4DFR. Education was based on the participant’s highest level of completed education defined as primary (<9 years), secondary (9 years), upper secondary (12 years) and university degree. Smoking status was obtained from the web-based lifestyle questionnaire and categorized as never-smoker and ex/current smoker.

### 2.6. Liquid Chromatography–Mass Spectrometry Analysis

Profiling of metabolites was performed in EDTA plasma samples using two liquid chromatography–mass spectrometry (LC-MS) methods, which have been described in more detail previously [20]. Briefly, proline betaine and β-carotene were measured in positive ion mode in samples separated on an Acquity UPLC BEH Amide column (1.7 µm, 2.1 × 100 mm; Waters Corporation, Milford, MA, USA). CMPF was measured in negative ion mode in samples separated on an ACE C18 column (1.7 µm; 2.1 × 100 mm; Advanced Chromatography Technologies Ltd., Aberdeen, UK). A more detailed description of the analytical procedures, data processing, normalization and metabolite identification is available in Appendix A: Method explanation [21] and Appendix A.

### 2.7. Statistical Analysis

The SPSS statistical computer package (version 24.0; IBM Corporation, Armonk, NY, USA) was used for all statistical analyses. Statistical significance was set at *p* < 0.05, and all *p*-vales are two-sided. The differences in baseline characteristics including dietary intakes from the 4DFR between the participants with a complete single dietary measurement compared to those with repeated dietary measurements were tested using general linear model for continuous variables, adjusted for age and sex where applicable, and chi-square test for categorical variables. Crude means and standard deviations for food intakes obtained from the 4DFR and the SFFQ, at baseline and at the second measurement, are presented in women and men separately. Regarding baseline measurements, data are presented both among all individuals (study sample for method comparisons) and among those with complete repeated measurements (study sample for reproducibility analyses). We used Spearman correlations (*rho = ρ*) because dietary intakes were not normally distributed. The correlation coefficients are presented stratified by sex because it is well known that both dietary habits and accuracy of dietary reporting could differ between women and men [22,23]. Spearman correlations were calculated to compare (i) intakes obtained from the 4DFR (g/d) and SFFQ (times/month and g/d for fish intake), (ii) intakes from repeated measurements and (iii) reported intakes and combined intake estimates of fatty fish, fruits and vegetables and citrus with objective biomarkers in plasma. Combined intake estimates were obtained by reducing reported intakes from the 4DFR and SFFQ at baseline into one score using principal component analysis, in line with previous combinations of reported intakes and biomarker levels [24].

Agreement of repeated intakes of nutrients and important food sources of fiber obtained using the 4DFR were also evaluated by cross-classification of intake quartiles and calculation of Cohen’s κ. We excluded participants reporting use of fish oil supplements when fish intake was compared to CMPF in plasma, and participants reporting use of multivitamin supplements when fruit and vegetable intake was compared to β-carotene in plasma. In addition to absolute intakes, energy adjusted intakes from the 4DFR were evaluated using intakes divided with non-alcohol energy intake.

## 3. Results

### 3.1. Baseline Characteristics and Reported Intakes from the Different Dietary Assessments

The participants with complete data from repeated dietary measurements (*n* = 180) were older and more frequently women compared to participants who only had complete dietary measurements from the baseline examination (*n* = 1421) (Table 1). In addition, those with repeated measurements had lower BMI, higher HDL-cholesterol and higher intake of polyunsaturated fat (PUFA) according to the 4DFRs, and a higher percentage among them had a university degree at baseline. There were no significant differences in intakes of energy, protein, carbohydrates, saturated fat, fiber, sucrose, meat, whole grain, fruit and vegetables or sugar-sweetened beverages between those with single and repeated dietary measurements.

Mean food intakes from the repeated 4DFRs and the repeated SFFQ are presented for both women and men in Appendix A. Means from baseline measurements are given both in the whole study sample and among those with repeated measurements. Mean nutrient intakes at baseline obtained from the 4DFR are presented in Appendix A, together with food intakes that were only reported in the 4DFR.

### 3.2. Comparison of Intakes Obtained from 4DFR and SFFQ

The median Spearman correlation between baseline food intakes assessed by the 4DFR and the SFFQ was 0.33 (range: 0.21–0.50) in the whole study sample (*n* = 1601), with the lowest correlation for cruciferous vegetables (i.e., cabbage, cauliflower, broccoli, Swedish turnip) and the highest for fruits and berries (Table 2). In sex-specific analysis, the lowest correlation was seen for cruciferous vegetables in men (ρ = 0.16), and the highest for low-calorie beverages in women (ρ = 0.52). Specific fruits, vegetables and fish showed lower correlation than the overall food groups. The correlations for citrus (ρ = 0.42) and berries (ρ = 0.34) were, for example, lower than that for total intakes of fruits and berries (ρ = 0.50), and correlations for fatty fish (ρ = 0.29) and lean fish (ρ = 0.26) were lower than that for total fish intake (ρ = 0.33).

We also examined correlations between baseline intakes obtained from the two methods restricted to those who participated in the repeated dietary measurements (*n* = 180). In that subsample, we observed slightly higher correlations between the two methods regarding most of the baseline intakes (median ρ = 0.39, range: 0.16–0.62 in analysis of women and men together) (Figure 2, Appendix A). Finally, the two methods were compared using mean intakes of the repeated measurements (2 × 4DFR vs. 2 × SFFQ). We observed higher correlation between the two methods for all intakes based on repeated measurements (median ρ = 0.44, range 0.26–0.74), compared to correlations between baseline measurements only, and especially regarding sub-groups of vegetables, soft bread and fatty fish (Figure 2, Appendix A). The median Spearman correlation between the two methods regarding the measurements performed 1.6 y after baseline was 0.35 (range: 0.28–0.68).

### 3.3. Reproducibility of Intakes Obtained from 4DFR

The median Spearman correlation between food and nutrient intakes obtained from the baseline 4DFRs and the repeated 4DFRs was 0.41 (range: 0.07–0.79) (Table 3). The correlations were in general higher for nutrients (median ρ = 0.48, range: 0.21–0.60) than for foods, with the lowest correlation observed for vitamin D and the highest for carbohydrates and water (Table 3). Correlations between nutrient intakes obtained from the repeated 4DFRs were in general somewhat higher in women; only correlations between intakes of PUFA (ρ = 0.24 vs. 0.37) and vitamin E (ρ = 0.30 vs. 0.48) indicated markedly lower correlations in women than in men. In men, the correlation between repeated β-carotene intake data was especially low (ρ = 0.05). Correlation between the repeated 4DFRs were, with a few exceptions, slightly higher for absolute intakes (median ρ = 0.48) than for energy-adjusted intakes (median ρ = 0.41) (data only shown for women and men together).

Regarding food intakes from the repeated 4DFRs, the Spearman correlations ranged between 0.06 (root vegetables in men) and 0.81 (coffee in women). In analysis of women and men together, the median Spearman correlation was 0.36 and we observed correlations of at least 0.45 for overall food groups such as total intakes of red meat, fruits, vegetables and dairy products. Lower correlations were in general observed for intakes of more specific foods. Correlations for specific vegetables (ρ = 0.21–0.30) were for example lower than the correlation between repeated measurements of total vegetable intake (ρ = 0.47). Similarly, the correlations for citrus (ρ = 0.39) and berries (ρ = 0.29) were lower than that for total intake of fruits and berries (ρ = 0.51), and correlations between repeated measurements of processed (ρ = 0.32) and unprocessed red meat (ρ = 0.33) were lower than that for total red meat (ρ = 0.47). Among examined dairy products, the lowest correlation was seen for cheese (ρ = 0.29) and the highest was seen for yoghurt/sour milk (ρ = 0.52), which was somewhat higher than that for total intake of dairy products (ρ = 0.45). In contrast to other overall food groups, total fish intake from the repeated 4DFRs showed a correlation of only ρ = 0.15 and specific intakes of fatty fish (ρ = 0.08) and lean fish (ρ = 0.07) showed even lower correlations. Correlations for fish intakes were weak in both genders.

Correlations between repeated measurements of vegetable intakes were found to be higher in women (ρ = 0.53 for total vegetable compared to ρ = 0.28 in men, and ρ = 0.22–0.41 for specific vegetables in women compared to ρ = 0.06–0.21 in men). The highest correlations between intakes from the repeated 4DFRs were seen for coffee and tea in both genders, with the highest correlation observed for coffee in women (ρ = 0.81) and for tea in men (ρ = 0.72).

On average 80% of the women were classified in the correct or adjacent quartile of the examined nutrient intakes from the repeated measurements, ranging from 70% for vitamin D to 90% for fiber (Table 4). In men, the corresponding average was somewhat lower (76%), ranging from 60% for β-carotene to 85% for monounsaturated fat and vitamin C (median = 77%). Kappa values were found to be ≥0.20 for most of the intakes. When specifically examining four important food sources of fiber (fruits and berries, vegetables, high-fiber bread and breakfast cereals/porridge), we observed similar results for the different sources, 79–82% of the women were found to be classified in the same or adjacent intake quartile of the different sources, and 68–74% of the men (Appendix A).

### 3.4. Reproducibility of Intakes Obtained from SFFQ

Regarding the selected foods included in the SFFQ, the median Spearman correlation between the repeated measurements was 0.59 (range: 0.32–0.79) (Table 5). The correlations for specific foods were in general in the same range as those for overall food groups. The Spearman correlation for fatty fish (ρ = 0.56 in analysis of women and men together) from the repeated SFFQs was for example similar to that for total fish intake (ρ = 0.54). The correlations for specific vegetables (ρ = 0.55–0.66) were similar to that for total vegetable intake (ρ = 0.58), and the correlations for citrus (ρ = 0.59) and berries (ρ = 0.69) were almost as high as that for total intakes of fruits and berries (ρ = 0.70). The lowest correlation between the repeated SFFQs was seen for butter for cooking in women (ρ = 0.29) and the highest was seen for fiber-rich crispbread in men (ρ = 0.80).

### 3.5. Validation of Fatty Fish Intake

Correlations between reported intakes of fatty fish and CMPF were higher for intakes from the SFFQ (ρ = 0.45 in women, ρ = 0.46 in men) than from the 4DFR (ρ = 0.28 in women, ρ = 0.22 in men) (Table 6). Correlations with CMPF did not improve when combining fatty fish intakes from the two dietary assessment methods using PCA. Correlations between the combined intake estimation and CMPF (ρ = 0.44 in women, ρ = 0.42 in men) were slightly lower than those observed for the SFFQ per se.

### 3.6. Validation of Citrus Intake

Total citrus intake from 4DFR showed higher Spearman correlations with proline betaine (ρ = 0.50 in women, ρ = 0.53 in men) than citrus intake from the SFFQ (intake from juice was not included in the SFFQ estimation) (ρ = 0.34 in women, ρ = 0.36 in men) (Table 6).

In men, the highest correlation with proline betaine was seen for citrus scores obtained when combining self-reported intakes from the two assessment methods (ρ = 0.55). In women, the correlation with proline betaine and the combined intake estimation was similar to that observed when using data from the 4DFR per se (ρ = 0.50).

### 3.7. Validation of Total Fruit and Vegetable Intake

Fruit and vegetable intake from the 4DFR (ρ = 0.35) and the SFFQ (ρ = 0.32) showed similar correlations with plasma concentration of β-carotene in analysis of women and men together (Table 6). In women, intakes from the SFFQ indicated somewhat lower correlation with the biomarker compared to intakes from the 4DFR. For both genders, highest correlations were seen between the combined intake estimation and β-carotene in plasma (ρ = 0.39 in analysis of men and women together).

## 4. Discussion

In this population-based Swedish cohort study, we observed moderate correlations between overall food groups in our main 4DFR method and an SFFQ. Higher agreement between the methods was seen when intake data from two time points were included, but the improvement varied between foods. Regarding the selected foods hypothesized to be insufficiently captured on a 4-day basis and therefore assessed by both methods, stronger correlations were seen between the repeated intakes obtained from the SFFQ data than between repeated 4DFR data. Regarding nutrients, agreement between intake levels from the repeated 4DFRs were found to be somewhat higher in women, where on average 80% were found to be classified into the correct or adjacent quartile. When validating intake data against objective plasma biomarkers, intake of fatty fish obtained from the SFFQ showed strongest correlation with CMPF, whereas a combined measure of fruit and vegetable intake obtained from the 4DFR and SFFQ showed stronger correlation with β-carotene, than intakes from either method per se. Combining the methods was also found to result in slightly higher correlation between intake data on citrus and the plasma biomarker proline betaine.

Both food records and food frequency questionnaires are prone to errors. However, correlation between repeated measurements showed higher overall precision of data obtained from the SFFQ compared to the 4DFR. In addition, validation of intakes against objective biomarkers indicated higher validity of intake data obtained from the SFFQ regarding fatty fish. On the other hand, the results indicated similar validity of intake data obtained from the 4DFR, compared to the SFFQ, regarding citrus intake and total fruit and vegetable intake, and that the best intake estimates could be obtained when combining those measures.

As the time between repeated measurements varies between studies, and as different studies did not evaluate reproducibility of identical food groups, comparison between studies is not straightforward. However, the reproducibility correlations of the repeated overall food group intakes obtained from 4DFRs in this study were in general moderate, although very weak for fish [25], whereas correlations between repeated intakes from the SFFQ were moderate or strong [25], and similar to those observed in other studies with FFQs [26,27].

Moreover, only fatty fish intake obtained from the SFFQ showed a correlation with the plasma biomarker CMPF that was in line with that observed in previous studies [9,28]. CMPF is incorporated in the cell membranes and is thereby a good marker of long-term fatty fish intake [29,30]. However, we cannot exclude that 4DFR data for fatty fish may be valuable when examining phenotypes that rapidly respond to dietary changes, such as gut bacterial composition. Our observed Spearman correlation coefficients of around 0.3 between reported intake of fruit and vegetables and plasma β-carotene, from the 4DFR as well as from the SFFQ, were similar to those observed in previous studies using different dietary assessment methods (0.17 to 0.46) [8,29,31,32], and among men we observed substantially stronger correlation with fruit and vegetable intake than when the same Riksmaten2010 4DFR was evaluated in another study population [33]. Proline betaine is an objective biomarker of citrus intake [14,15], and we observed correlations with total citrus intake from the 4DFR from ρ = 0.50, which is comparable to [34,35] or somewhat stronger than [30] those reported in other studies. Our lower correlation coefficients regarding citrus intakes obtained from the SFFQ were probably due to the fact that the questionnaire did not include juice intake, and citrus juice could be considered as an important source of proline betaine.

To improve dietary data quality, our observed correlations between reported intakes and biomarkers indicate that combining estimates from the 4DFR and SFFQ may result in slightly better estimations of true habitual intakes regarding some foods. These findings are important to consider when designing future dietary assessment studies. However, our biomarker validation does not suggest that 4DFRs contribute importantly to valid estimations of habitual fatty fish intake. Instead, the observed markedly higher correlation between the 4DFR and SFFQ using mean intakes of baseline and repeated measurements of fatty fish intake (8 d = 2 × 4 d) compared to correlations between baseline measurements only indicate that repeating the measurements in all individuals, at another point in time, could improve the quality of estimated habitual fatty fish intake. In fact, repeated 4DFRs may give a better estimate of usual long-term intakes of fatty fish, bread and different types of vegetables compared to the single 4DFRs. Finally, in addition to repeated measurements and combined dietary assessment methods, a third opportunity might be to also include biomarker data and thereby take advantage of the strengths of both the 4DFR and the SFFQ, as well as objective intake estimates [7]. This possibility could be evaluated by examining combined intake estimates in relation to markers of chronic disease in a future study. However, it is important to consider the additional costs and representativeness of the individuals in the study sample that agreed to participate in such combined measurements in a large study population.

The strengths of this study include the large sample with intake data from both the 4DFR and the SFFQ. In addition, data is available from repeated measurements and objective plasma biomarkers regarding specific intakes. This enables comparison and evaluation of different aspects of data quality of importance when selecting and combining different types of data for differing purposes, such as studies of long-term diet in relation to disease development or current diet in relation to gut bacterial composition. A limitation of the study is that none of our data can be regarded as a golden standard, as both the 4DFR and the SFFQ are subject to different types of systematic errors. Consequently, our method comparisons do not give any strong general guidance regarding reported intakes in relation to true usual intakes. However, although objective biomarkers also have errors, comparisons against objective biomarkers showed correlations in line with those of previous studies. Unfortunately, we do not have repeated biomarker data. Furthermore, we cannot guarantee that the sample with complete repeated dietary measurements is perfectly representative of the whole study sample with regard to accuracy of dietary reporting, because those with repeated data may be more health conscious; they were more often women and normal weight, and they had higher education, higher HDL-cholesterol and higher intake of PUFA. To enable comparison of the two different dietary assessment methods regarding their reproducibility, we only included individuals with repeated data from both methods. We therefore ended up with a small sample of men (*n* = 65) included in the reproducibility study, which may explain some of the rather weak correlations between some of the specific food intakes obtained from the 4DFRs. On the other hand, we did not observe stronger correlations between repeated 4DFRs, when adding 130 individuals with repeated dietary data restricted to the 4DFR (range: ρ = 0.01 for fatty fish to ρ = 0.66 for coffee). Moreover, due to the small study sample, we could not adjust the reproducibility correlations between repeated 4DFRs for season and weekday. However, adjustment for those factors, in future diet-disease studies, may improve observed risk estimates. Finally, it is worth mentioning that, as diet varies over time, the correlation between repeated measurements is influenced not merely by the precision of the methods, but also by true dietary change over time. On the other hand, both factors are of importance when aiming to assess long-term diet.

## 5. Conclusions

Regarding overall food groups, moderate correlations were in general seen between two dietary assessment methods and between repeated measurements. Our findings also showed that long-term intake of irregularly consumed foods was more accurately captured by the SFFQ compared to a single 4DFR and that data could be improved by repeated measurements. Assessment of fatty fish intake by the SFFQ indicated more valid estimations compared to fish intake from the 4DFR, whereas a combined measure from both diet assessment methods indicated most optimal estimations of fruit and vegetable intakes. These findings will provide guidance for how dietary data from the MOS cohort can be used and combined in future studies.

## Figures and Tables

**Figure 1 nutrients-13-01579-f001:**
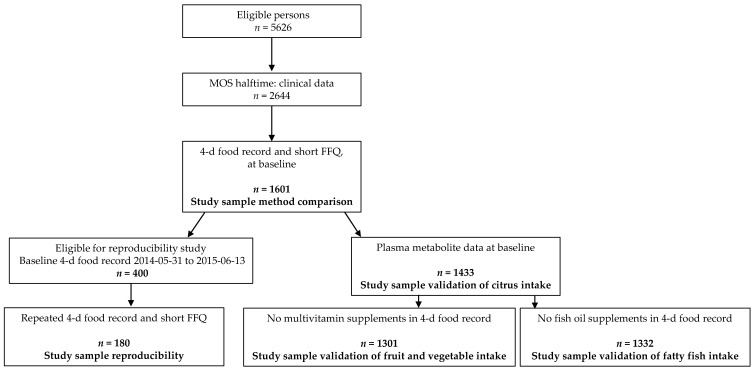
Flowchart of study samples in the Malmö Offspring Study until end of April 2017.

**Figure 2 nutrients-13-01579-f002:**
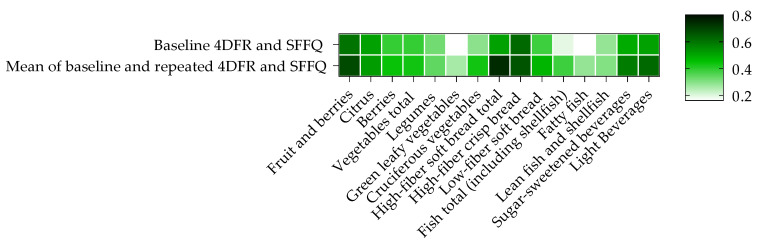
Correlations between 4DFR and SFFQ based on baseline vs. mean of baseline and repeated measurements. Stronger correlations were observed between the two methods for all intakes based on mean of baseline and repeated measurements (2 × 4DFR and 2 × SFFQ) compared to correlations between baseline measurements only, especially regarding specific vegetables, soft bread and fatty fish. Data from individuals with repeated dietary measurements in the Malmö Offspring Study (*n* = 180).

**Table 1 nutrients-13-01579-t001:** Baseline characteristics among the 1601 MOS participants with single (*n* = 1421) or repeated dietary data from both 4-d food records (4DFR) and the short food frequency questionnaire (SFFQ) (*n* = 180).

Baseline Characteristics ^a^	Participants with Only Baseline Diet Data (*n* = 1421)	Participants with Repeated Diet Data (*n* = 180)	*p* Value ^b^
Age (y)	40.3 (39.6, 41.0)	46.2 (44.2, 48.2)	<0.001
Sex (women *n* (%))	770 (54.2)	115 (63.9)	0.01
BMI (kg/m^2^)	25.8 (25.6, 26.1)	24.8 (24.1, 25.4)	0.003
Systolic blood pressure (mmHg)	116.5 (115.9, 117.2)	114.9 (113.0, 116.7)	0.09
Diastolic blood pressure (mmHg)	71.7 (71.3, 72.1)	70.5 (69.3, 71.7)	0.07
Fasting glucose (mmol/L)	5.5 (5.4, 5.5)	5.4 (5.3, 5.6)	0.33
Triglycerides (mmol/L)	1.1 (1.1–1.2)	1.0 (0.9–1.1)	0.09
HDL-C (mmol/L)	1.61 (1.59, 1.63)	1.69 (1.63, 1.75)	0.02
LDL-C (mmol/L)	3.17 (3.12, 3.21)	3.10 (2.97, 3.23)	0.32
Total cholesterol (mmol/L)	4.97 (4.91, 5.01)	4.92 (4.78, 5.07)	0.58
Total energy (kcal/d)	2028 (1998, 2058)	2070 (1984, 2155)	0.37
Protein (E%)	17.6 (17.4, 17.8)	17.3 (16.7, 17.9)	0.30
Carbohydrates (E%)	45.1 (44.7, 45.5)	45.1 (44.0, 46.2)	0.97
Fat (E%)	37.3 (36.9, 37.6)	37.6 (36.5, 38.6)	0.60
Saturated fat (E%)	14.2 (14.0, 14.4)	13.9 (13.4, 14.4)	0.32
PUFA (E%)	6.0 (5.9, 6.1)	6.4 (6.1, 6.7)	0.02
Fiber (g/1000kcal)	9.7 (9.6, 9.9)	9.9 (9.4, 10.3)	0.59
Sucrose (E%)	8.4 (8.2, 8.7)	8.3 (7.6, 8.9)	0.66
Alcohol (g/d)	14.0 (13.1, 14.9)	14.9 (12.3. 17.5)	0.52
Red meat (g/d)	87.1 (84.2, 89.0)	85.0 (76.9, 93.1)	0.63
Fruits and vegetables (g/d)	264.8 (256.4, 273.1)	257.0 (233.2, 280.8)	0.55
Whole grain (g/d)	35.2 (33.1, 37.3)	35.4 (29.4, 41.3)	0.96
Sugar-sweetened beverages (g/d)	94.4 (86.2, 102.6)	85.6 (62.4, 108.9)	0.48
Physical activity (PAL)	1.66 (1.66, 1.67)	1.66 (1.64, 1.68)	0.43
Smokers, ex or current (*n* (%))	500 (37.3)	63 (35.4)	0.62
Higher education, university degree (*n* (%))	517 (38.7)	90 (51.1)	0.01

BMI: body mass index, HDL-C: high density lipoprotein cholesterol; LDL-C: low density lipoprotein cholesterol; PUFA: polyunsaturated fat. ^a^ Information was missing for some participants: SBP (*n* = 1402/*n* = 179); f-glucose (*n* = 1420/*n* = 180); Triglycerides (*n* = 1406/*n* = 180); HDL-C and total cholesterol (*n* = 1417/*n* = 180); LDL-C (*n* = 1416/*n* = 180); Education (*n* = 1337/*n* = 176); Smoking (*n* = 1341/*n* = 178); ^b^ The general linear model, adjusted for age and sex when applicable, for continuous and chi2-test for categorical variables. Mean (±SD) for continuous and *n* (%) for categorical variables. *p* < 0.05. Dietary data from 4-d food records.

**Table 2 nutrients-13-01579-t002:** Spearman correlations * between food intakes assessed by the 4-d food record (4DFR) (g/d) and the short food frequency questionnaire (SFFQ) (times/month and g/d for fish intake), in 1601 women and men from the Malmö Offspring Study.

Dietary Factor	ρ Baseline Measurements All (*n* = 1601)	ρ Baseline Measurements Women (*n* = 885)	ρ Baseline Measurements Men (*n* = 716)
Fruit and berries	0.50	0.48	0.45
Citrus	0.42	0.43	0.39
Berries	0.34	0.33	0.30
Vegetables total	0.35	0.35	0.35
Legumes	0.26	0.30	0.21
Green leafy vegetables	0.31	0.31	0.28
Cruciferous vegetables	0.21	0.24	0.16
High-fiber soft bread total	0.33	0.31	0.36
High-fiber crisp bread	0.35	0.35	0.31
Low-fiber soft bread	0.32	0.34	0.27
Fish total (including shellfish)	0.33	0.31	0.35
Fatty fish	0.26	0.28	0.26
Lean fish and shellfish	0.26	0.25	0.29
Sugar-sweetened beverages	0.42	0.39	0.44
Low-calorie beverages	0.49	0.52	0.44

* *p* < 0.01 for all correlations.

**Table 3 nutrients-13-01579-t003:** Spearman correlations between dietary intakes from the first and repeated 4-d food record (4DFR) (g/d) in the Malmö Offspring Study (*n* = 180).

Dietary Intakes	ρ All *n* = 180	ρ Women *n* = 115	ρ Men *n* = 65	ρ Eneradj ^a^ All *n* = 180
Energy	0.51 *	0.57 *	0.43 *	
Carbohydrates (non fiber)	0.60 *	0.62 *	0.53 *	0.54 *
Fat	0.43 *	0.45 *	0.38 *	0.40 *
Saturated fat	0.39 *	0.44 *	0.28 *	0.34 *
Monounsaturated fat	0.44 *	0.42 *	0.42 *	0.37 *
Polyunsaturated fat	0.29 *	0.24 *	0.37 *	0.21 *
Protein	0.52 *	0.47 *	0.47 *	0.51 *
Fiber	0.58 *	0.68 *	0.36 *	0.58 *
Sucrose	0.41 *	0.43 *	0.36 *	0.32 *
Monosaccharides	0.53 *	0.58 *	0.48 *	0.50 *
Disaccharides	0.47 *	0.50 *	0.44 *	0.41 *
Vitamin C	0.49 *	0.46 *	0.47 *	0.52 *
Folate	0.48 *	0.54 *	0.39 *	0.50 *
Retinol equivalent	0.35 *	0.36 *	0.33 *	0.34 *
β-carotene	0.38 *	0.55 *	0.05	0.41 *
Vitamin D	0.21 *	0.17	0.28 *	0.20 *
Vitamin E	0.36 *	0.30 *	0.48 *	0.40 *
Alcohol	0.51 *	0.52 *	0.46 *	0.47 *
Iron	0.48 *	0.54 *	0.33 *	0.46 *
Zink	0.49 *	0.43 *	0.46 *	0.31 *
Magnesium	0.55 *	0.60 *	0.46 *	0.48 *
Calcium	0.43 *	0.52 *	0.29 *	0.42 *
Sodium	0.49 *	0.43 *	0.44 *	0.32 *
Water (in beverages and food moisture)	0.60 *	0.62 *	0.57 *	0.48 *
Whole grain	0.37 *	0.38 *	0.34 *	0.40 *
Low-fiber Soft bread total	0.36 *	0.40 *	0.25 *	0.33 *
High-fiber soft bread total	0.36 *	0.38 *	0.41 *	0.43 *
High-fiber crisp bread	0.32 *	0.35 *	0.34 *	0.34 *
Breakfast cereals/porridge	0.51 *	0.53 *	0.50 *	0.50 *
Rice, pasta and other grains	0.28 *	0.20 *	0.43 *	0.22 *
Nuts/seeds	0.40 *	0.47 *	0.15	0.40 *
Red meat, non processed	0.33 *	0.30 *	0.26 *	0.28 *
Processed meat	0.32 *	0.31 *	0.19	0.27 *
Total red meat	0.47 *	0.42 *	0.40 *	0.43 *
Poultry	0.21 *	0.24 *	0.16	0.25 *
Vegetarian products ^b^	0.43 *	0.42 *	0.51 *	0.44 *
Egg	0.29 *	0.31 *	0.26 *	0.30 *
Total dairy	0.45 *	0.38 *	0.56 *	0.42 *
Yoghurt/sour milk	0.52 *	0.54 *	0.45 *	0.54 *
Milk, non fermented total	0.47 *	0.50 *	0.44 *	0.43 *
Cheese	0.29 *	0.33 *	0.21 *	0.30 *
Butter based spreads	0.44 *	0.52 *	0.30 *	0.45 *
Oil-based spreads	0.48 *	0.49 *	0.43 *	0.48 *
Fatty fish	0.08	0.07	0.09	0.05
Lean fish and shellfish	0.07	0.07	0.06	0.11
Fish total	0.15 *	0.15	0.16	0.22 *
Vegetables total	0.47 *	0.53 *	0.28 *	0.51 *
Legumes	0.23 *	0.26 *	0.16	0.23 *
Root vegetables	0.27 *	0.41 *	0.06	0.27 *
Green leafy vegetables	0.30 *	0.34 *	0.21	0.32 *
Cruciferous vegetables	0.21 *	0.22 *	0.15	0.20 *
Potatoes	0.37 *	0.34 *	0.38 *	0.36 *
Fruit and berries, total	0.51 *	0.46 *	0.38 *	0.38 *
Citrus	0.39 *	0.29 *	0.27 *	0.32 *
Berries	0.29 *	0.31 *	0.16	0.30 *
Sweets/pastry/desserts	0.32 *	0.19 *	0.48 *	0.32 *
Jam, sugar and honey	0.21 *	0.24 *	0.14	0.20 *
Salty snacks	0.31 *	0.24 *	0.44 *	0.31 *
Food replacement products	0.44 *	0.41 *	0.49 *	0.43 *
Sugar-sweetened beverages	0.43 *	0.33 *	0.53 *	0.42 *
Low-calorie beverages	0.47 *	0.31 *	0.67 *	0.46 *
Juice	0.34 *	0.32 *	0.34 *	0.32 *
Tea	0.69 *	0.67 *	0.72 *	0.70 *
Coffee	0.79 *	0.81 *	0.75 *	0.79 *
Water (tap and bottled)	0.63 *	0.60 *	0.58 *	0.62 *

* *p* < 0.01 for indicated correlations. ^a^ Energy adjusted dietary intakes in the 180 participants were calculated using intakes divided with non-alcohol energy intake; ^b^ Meat/milk/cheese replacement products.

**Table 4 nutrients-13-01579-t004:** Agreement between quartiles of nutrient intakes from the first and repeated 4-d food record (4DFR) in the Malmö Offspring Study (*n* = 180).

	Women Cross-Classification (%)				Men Cross-Classification (%)				All
Dietary Intakes	Perfect Agreement (Same Quartile)	Same or Adjacent Quartile	Gross Misclassification (Opposite Quartile)	Κ	Perfect Agreement (Same Quartile)	Same or Adjacent Quartile	Gross misclassification (Opposite Quartile)	Κ	K
Energy	47.0	84.4	3.4	0.28	38.5	76.8	3.1	0.13	0.25
Carbohydrates (non fiber)	49.5	86.1	1.7	0.32	44.6	80.0	4.6	0.23	0.30
Fat	38.4	80.9	5.2	0.17	44.7	75.3	7.7	0.25	0.21
Saturated fat	40.0	75.8	3.4	0.20	40.1	67.6	7.7	0.20	0.20
Monounsaturated fat	35.7	76.6	5.2	0.14	36.9	84.8	9.2	0.15	0.15
Polyunsaturated fat	33.9	71.3	7.8	0.12	44.6	80.0	7.7	0.27	0.17
Protein	42.7	80.8	3.5	0.14	38.5	80.0	4.6	0.22	0.22
Fiber	48.7	89.6	1.8	0.31	32.4	81.6	4.6	0.13	0.25
Sucrose	40.1	79.1	5.2	0.23	43.1	72.3	9.2	0.20	0.22
Alcohol	40.9	79.9	2.6	0.20	44.6	76.9	10.8	0.25	0.23
Vitamin C	45.2	80.1	2.6	0.27	46.1	84.7	9.2	0.27	0.27
Folate	40.9	79.1	1.8	0.21	38.5	72.3	3.1	0.19	0.20
β-carotene	40.0	81.0	2.6	0.20	29.3	60.1	6.1	0.04	0.15
Vitamin D	33.0	70.3	10.5	0.11	27.7	73.6	11.0	0.04	0.08
Vitamin E	32.2	70.4	6.1	0.09	46.2	76.9	1.5	0.28	0.16
Iron	47.0	83.5	3.4	0.29	40.1	72.4	1.5	0.18	0.26
Zink	45.3	77.4	4.4	0.26	46.2	81.5	4.6	0.26	0.27
Magnesium	43.6	86.2	1.7	0.25	41.6	81.5	7.7	0.22	0.24
Calcium	38.2	80.9	2.6	0.18	27.7	70.7	6.2	0.03	0.13
Sodium	44.3	80.9	4.3	0.24	43.2	71.6	6.1	0.19	0.25
Water	51.2	84.3	1.8	0.35	39.9	81.7	1.5	0.20	0.30

**Table 5 nutrients-13-01579-t005:** Spearman correlations * between repeated assessments of food intakes using the short food frequency questionnaire (SFFQ) (times/month and g/d for fish intake) in 180 women and men from the Malmö Offspring Study.

Dietary Factor	ρ All	ρ Women	ρ Men
Low-fiber soft bread total	0.70	0.67	0.68
Low-fiber crispbread	0.40	0.38	0.45
High-fiber soft bread total	0.73	0.69	0.79
Medium high-fiber soft bread	0.60	0.56	0.63
Very high-fiber soft bread	0.61	0.58	0.66
High-fiber crisp bread	0.66	0.58	0.80
Fish total	0.54	0.58	0.44
Fatty fish	0.56	0.61	0.50
Lean fish and shellfish	0.55	0.51	0.62
Fish products times	0.48	0.51	0.45
Vegetables total	0.58	0.57	0.61
Legumes	0.61	0.60	0.63
Green leafy vegetables	0.55	0.64	0.37
Cruciferous vegetables	0.57	0.54	0.56
Onions	0.66	0.71	0.57
Tomatoes	0.60	0.64	0.51
Carrots	0.59	0.70	0.42
Other vegetables	0.48	0.47	0.51
Fruit and berries total	0.70	0.66	0.72
Fruits total	0.71	0.66	0.71
Citrus	0.59	0.57	0.63
Other fruits	0.64	0.64	0.53
Berries	0.69	0.72	0.61
Sugar-sweetened beverages	0.74	0.68	0.76
Low-calorie beverages	0.68	0.70	0.69
Energy/sport beverages	0.58	0.51	0.65
Butter for cooking	0.40	0.29	0.59
Margarine for cooking	0.44	0.42	0.47
Oil/liquid margarine for cooking	0.57	0.53	0.65
Oil/vinaigrette on salad	0.60	0.61	0.60
Energy bars/protein powder	0.58	0.62	0.56
Protein beverages	0.41	0.29	0.56
Food replacement products	0.32	0.28	0.39
Probiotic products	0.44	0.52	0.26
Home cooked meals	0.71	0.72	0.69
Precooked/ready to eat dishes	0.51	0.53	0.48
Eating out at restaurants	0.79	0.79	0.76
Take-away/fast food	0.72	0.76	0.66

* *p* < 0.01 for all correlations.

**Table 6 nutrients-13-01579-t006:** Spearman correlations * between fatty fish, citrus and fruits and vegetable intake estimations and the plasma biomarkers 3-carboxy-4-methyl-5-propyl-2-furanpropanoic acid (CMPF), proline betaine, and β-carotene in the Malmö Offspring Study.

*Fatty Fish/CMPF*	*n*	4DFR	SFFQ	Combination 4DFR and SFFQ by PCA
All	1332 ^a^	0.25	0.46	0.43
Women	731	0.28	0.45	0.44
Men	601	0.22	0.46	0.42
*Citrus*/ *Proline betaine*				
All	1433	0.51	0.35	0.53
Women	794	0.50	0.34	0.50
Men	639	0.53	0.36	0.55
*Fruits vegetable*/ *β-carotene*				
All	1301 ^b^	0.35	0.32	0.39
Women	713	0.34	0.27	0.35
Men	588	0.30	0.30	0.36

CMPF: 3-carboxy-4-methyl-5-propyl-2-furanpropanoic acid; 4DFR: 4-d food record; SFFQ: short food frequency questionnaire; PCA: principal component analysis. ^a^ in non-users of fish oil supplements; ^b^ in non-users of multivitamin supplements. * *p* < 0.01 for all correlations.

## Data Availability

The data presented in this study are available upon request from the corresponding author. The data are not publicly available due to privacy and ethnical reasons.

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
