# Peer review of "Dietary Data in the Malmö Offspring Study–Reproducibility, Method Comparison and Validation against Objective Biomarkers"

_nutrients, 2021, doi:10.3390/nu13051579_

Round 1

Reviewer 1 Report

This manuscript presents some data obtained in the Malmö study, comparing intakes from four-day food dietary records (4DFR), a semiquantitative short FFQ and data on biomarkers (CPMF, betacarotene and proline betaine). The manuscript is well structured, and wording is adequate. The study is of obvious interest for Nutrients readers.

After a carefully reading of the paper, these are my commentaries and suggestions:

-  The authors do not mention whether the semiquantitative SFFQ had been previously validated or not. Please include information about it.

- I understand the Spearman correlation coefficient are indicated to study the association between 4DFR and SFFQ estimates of food consumption and nutrient intakes, and also between any of these both methods and biomarkers, because the variables tend to change at the same time, but not necessarily at a constant rate. But I don’t understand why Spearman coefficient has been used to study association between repeated 4DFR and SSFQ. Moreover, I think it is incorrect because, theoretically, changes between both variables should be at constant rate. Can you please provide the reasons that support the decision you made? If not, I suggest you use Pearson's coefficient.

Reviewer 2 Report

This paper seeks to evaluate performance of two dietary assessment tools, a 4-day food record and a food frequency questionnaire among Swedish adults. The evaluation includes both objective validation against specific biomarkers, relative validation comparing the two methods against each other and finally reproducibility comparing two assessments of the same method – for both tools. Thus, the authors present state of the art validation of dietary assessment tools and a very comprehensive evaluation.

The paper shows important results when it comes to designing future dietary assessment studies. It seems that recommending combining methods could be beneficial depending on which foods or nutrients are under investigation. This is an important point compared to more traditional thinking of either/or which could be emphasized a bit more in the discussion but still taking the representativeness into consideration. On the other hand, combination of methods may also be very study specific considering that in large cohort studies, repeated daily assessment may be too expensive.

The other questions that came to my mind while reading the manuscript was sufficiently answered in the discussion. I have only a few minor reflections on this paper.

Have the authors tested interaction or how was it decided to adjust/stratify by sex? Please specify rationale in methods for stratification in specific situations vs adjustment.

Have the authors considered multiple testing? The paper holds an enormous number of correlations. Please add a few comments in the discussion if you agree.

Have the authors considered other ways to sum up/shorten tables and results? The one figure showing correlations is a nice example, could this be somehow applied other places as well?

Table 1, smokers and education does not seem to be % only. Check symbols in list of characteristics.

Round 2

Reviewer 1 Report

The manuscript have been significantly improved. Congratulations.